



# Middle Miocene climate evolution in the Northern Mediterranean region (Digne-Valensole basin, SE France)

Armelle Ballian[1,2], Maud J.M. Meijers[1,3], Isabelle Cojan[4], Damien Huyghe[4], Miguel Bernecker[2], Katharina Methner[1,5], Mattia Tagliavento[2,6], Jens Fiebig[2], Andreas Mulch[1,2]

[1]Senckenberg Biodiversity and Climate Research Centre (SBiK-F), Frankfurt am Main, Germany

[2]Goethe University Frankfurt, Institute of Geosciences, Frankfurt am Main, Germany

[3]University of Graz, Department of Earth Sciences, NAWI Graz Geocenter, Graz, Austria

[4]PSL University, MinesParis, Centre de Géosciences, Fontainebleau, France

[5]Institute for Earth System Science and Remote Sensing, University of Leipzig, Germany

[6]Division of Geological and Planetary Sciences, California Institute of Technology, Pasadena, CA, USA

*Correspondence to:* Armelle Ballian (armelle.ballian@senckenberg.de)

**Abstract.** During the Middle Miocene, the Earth shifted from a warm state, the Miocene Climatic Optimum (MCO, 16.9–14.7 Ma), to a colder state associated with the formation of extensive and permanent ice sheets on Antarctica. This climatic shift, the Middle Miocene Climatic Transition (MMCT, 14.7–13.8 Ma) strongly affected the composition and structure of major biomes, ocean circulation, as well as precipitation patterns. Although Middle Miocene climate dynamics are well documented in marine records, our knowledge of terrestrial climate

change is not well constrained. Here we present a long-term (23–13 Ma) stable ($\delta^{13}$C, $\delta^{18}$O) and clumped ($\Delta_{47}$) isotope record of soil carbonates from a northern Mediterranean Alpine foreland basin (Digne-Valensole Basin, France). $\Delta_{47}$-derived soil carbonate formation temperatures indicate a highly dynamic dry season temperature pattern that is consistent with multiple periods of reorganization of atmospheric circulation during the MCO. We propose that changes in atmospheric circulation patterns modified the seasonality of precipitation and, ultimately,

the timing of pedogenic carbonate formation. Consequently, $\Delta_{47}$ soil carbonate temperature data record the combined effects of long-term regional temperature and carbonate formation seasonality change. The data are consistent with the existence of a proto-Mediterranean climate already during certain MCO time intervals. Following the MMCT, the stable and clumped isotope record displays pronounced cooling after 13.8 Ma accompanied by a rather large (-5.0 ‰) decrease in soil water $\delta^{18}$O values. Our northern Mediterranean foreland

basin climate record shares strong similarities with time-equivalent records from the terrestrial European mid-latitudes and the global oceans and enhances our understanding of the circum-Alpine Middle Miocene terrestrial climate dynamics.



## 1. Introduction

The Miocene Climatic Optimum (MCO, 16.9–14.7 Ma) represents a warm but dynamic climatic interval and was followed by a transition to a colder state, the Middle Miocene Climatic Transition (MMCT, 14.7–13.8 Ma). The MMCT was associated with the inception of major, permanent Antarctic ice sheets (e.g., Flower and Kennett, 1994; Holbourn et al., 2005) and is considered to prelude one of the major cooling steps in Earth's Cenozoic climate history (Flower and Kennett, 1994; Holbourn et al., 2007). Global cooling had profound impacts on the distribution of major biomes (Jiménez-Moreno and Suc, 2007), global ocean circulation and ice sheet dynamics (e.g., Holbourn et al., 2014; Zachos et al., 2001), as well as continental temperature and precipitation patterns (e.g., Methner et al., 2020; Pound et al., 2012).

The MCO corresponds to one of the warmest intervals of the Neogene period (Flower and Kennett, 1994; Holbourn et al., 2022; Zachos et al., 2001). With higher global temperatures and moderately higher atmospheric $CO_2$ concentrations (~400-600 ppm; Steinthorsdottir et al., 2021) than today it bears characteristics of future global warming scenarios (Longman et al., 2022; Steinthorsdottir et al., 2021b). Refining the understanding of climate variability during the MCO may, therefore, underpin projections of anthropogenic climate change in particular for the (circum-)Mediterranean region, which has suggested to become highly vulnerable to intensified future droughts (e.g., Lionello, 2012).

Despite the intense study of Earth's Miocene climate history, our knowledge of Middle Miocene continental climate dynamics remains fragmentary. Efforts in reconstructing European continental climate conditions during the Middle Miocene focused on Central (e.g., Jiménez-Moreno et al., 2008; Methner et al., 2020; Mosbrugger et al., 2005) or Southern Europe (e.g., Abels et al., 2009; Jiménez-Moreno and Suc, 2007) and highlight possible latitudinal differences in (hydro-)climate (Fauquette et al., 2007; Jiménez-Moreno and Suc, 2007).

Well-dated paleosol sequences provide excellent archives to study long-term continental (hydro-)climatic changes (e.g., Quade et al., 2007), soil formation (Sheldon and Tabor, 2009) and soil water characteristics within soil profiles (Zamanian et al., 2021). They also allow to estimate climate parameters such as mean annual precipitation (MAP) and mean annual temperature (MAT) (Gillot et al., 2022; Retallack, 2005), and soil temperatures during pedogenic carbonate formation through clumped isotope thermometry (e.g., Kelson et al., 2018; Meijer et al., 2024; Methner et al., 2020). Clumped isotope ($\Delta_{47}$) thermometry allows to reconstruct the formation temperature of (pedogenic) carbonate, independent of the oxygen isotopic composition of the parental fluid (Eiler, 2011). By simultaneously obtaining carbon ($\delta^{13}C$) and oxygen ($\delta^{18}O$) isotopic compositions in addition to $\Delta_{47}$ values of pedogenic carbonates, it is possible to constrain the oxygen isotopic composition of soil water ($\delta^{18}O_w$) assuming equilibrium fractionation (e.g., Kim et al., 2007). This approach was successfully applied to paleosols to document long-term continental temperature change as well as temperature seasonality and changes in moisture sources (e.g., Kelson et al., 2018; Meijer et al., 2024; Methner et al., 2020; Rugenstein et al., 2022). Embedded within a solid geochronological framework, such paleosol records in terrestrial basins may hence provide continental paleoclimatic information with a temporal resolution that permits direct comparison with marine proxy records (e.g., Methner et al., 2020).

Here, we present $\delta^{18}O$, $\delta^{13}C$ and $\Delta_{47}$ data from ca. 23–13 Ma old pedogenic carbonate nodules of the Digne-Valensole basin (DVB, France, Northern Mediterranean region). Today, the DVB is located at an intermediate





latitude between the well-studied Middle Miocene localities in Southern (Madrid and Calatayud-Daroca basins, Spain; e.g., Abels et al., 2009) and Central Europe (e.g., Northern Alpine Foreland Basin, NAFB, Switzerland;
Kälin and Kempf, 2009; Methner et al., 2020).

**2. Geological and climatic settings**

**2.1   Geology of the Digne-Valensole basin**

At the intersection of Mediterranean, semi-continental, and mountainous climates, the modern DVB - as part of the western Alpine foreland basin - is characterized by an altered Mediterranean climate ('climat méditerranéen
altéré', Joly et al., 2010). The latter differs from the typical Mediterranean climate by a higher frequency of "cold" days and a greater annual precipitation amount (800-950 mm) distributed non-uniformly across the year (Joly et al., 2010).

Mediterranean climate is defined as semi-arid, with dry and hot summers that contrast mild and wet winters. During the winter months, cool and wet air masses from the North Atlantic Ocean deliver moisture to the western
Alpine foreland, including the DVB (e.g., Seager et al., 2019). During the summer months, a shift in prevailing wind patterns leads to progressively drier conditions. Moisture from the Mediterranean Sea is episodically transported northward to the DVB region by southerly-southwesterly winds. The Eocene to Pliocene Digne-Valensole basin represents a deformed segment of the Alpine foreland basin system that extends along a narrow SW-NE trending belt west of the Western Alps (Fig. 1A-B). The DVB molasse currently covers ca. 1500 km$^2$
(Fig. 1A) and is bordered by the Digne thrust front in the north and east and the Durance strike-slip fault in the west ( Fig. 1B; Ford et al., 1999).

Combined advancement of the Alpine orogenic front and variations in sea level (Bauer, 2006; Bialkowski, 2002; Cojan et al., 2013) affected the Late Eocene to Pliocene basin-fill. The overall ca. 3000 m thick Tertiary basin succession (the 'Digne Molasse'; Cojan et al., 2013 and ref. therein; Gigot et al., 1974) is composed of alternating
continental to shallow marine deposits with lateral variations in facies and stratigraphic thickness, as well as internal unconformities (Gigot et al., 1974). The Digne Molasse (Gigot et al., 1974) is divided into four formations, which include from base to top: (1) the Chattian Molasse Rouge (thickness of up to 500 m), which is characterized by alluvial and fluvial facies; (2) the Chattian−Burdigalian Molasse Grise or Molasse Intermédiaire (thickness of several hundred meters), which consists of deltaic and distal floodplain deposits; (3) the Burdigalian-Tortonian
Molasse Marine (thickness up to 1600 m), dominated by tidal and coastal deposits and characterized by several transgression-regression cycles; and (4) the Mio-Pliocene Molasse Jaune (thickness up to 300 m), made up of coarse-grained alluvial sedimentary rocks. Most of the material in this study originates from soil horizons within the Molasse Grise and Molasse Marine formations, as well as a few samples from the Molasse Rouge formation.



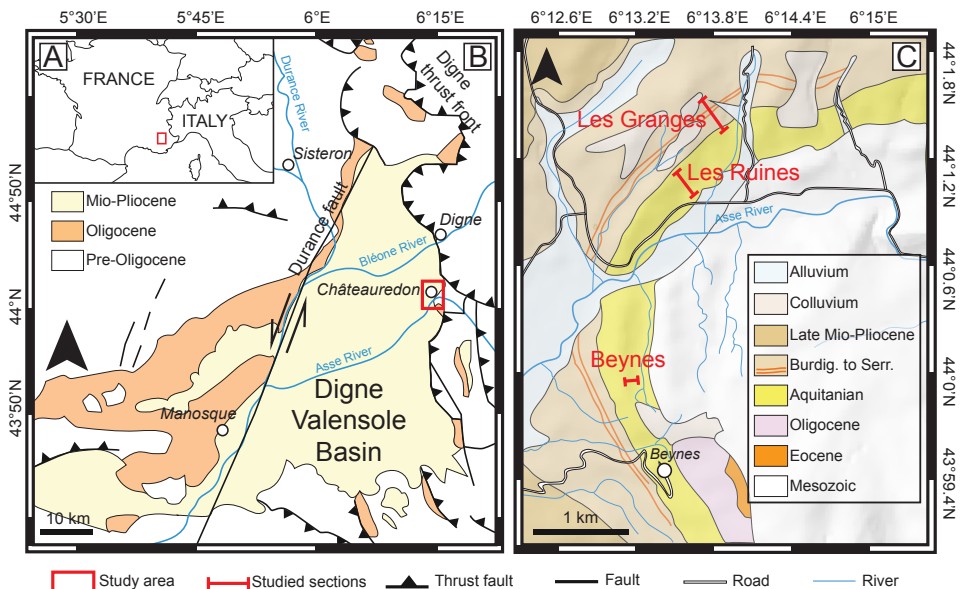

**Figure 1:** (A) Map showing South-central Europe. Location of the Digne-Valensole basin (DVB) in SE France (Fig. 1B) is indicated by the red rectangle. (B) Geological map of the DVB and study area (red rectangle, Fig. 1C) at the margin of the Digne thrust front (after Gillot et al., 2022). (C) Locations of Les Granges, Les Ruines, and Beynes sections within the DVB (after Cojan and Gillot, 2022).

## 2.1 Middle Miocene climate variability

The Miocene (23.03–5.33 Ma) period is characterized by major environmental change including cool periods associated with glaciation of Antarctica and the northern hemisphere that were interrupted by warm events, such as the Miocene Climatic Optimum (MCO, 16.9–14.7 Ma; Holbourn et al., 2014, 2015; Jiménez-Moreno et al., 2005; Mudelsee et al., 2014). Well-dated oceanic records provide numerous records of Miocene climate (e.g., Flower and Kennett, 1994; Holbourn et al., 2014, 2015; Mudelsee et al., 2014; Westerhold et al., 2020; Zachos et al., 2001). Following protracted Cenozoic cooling, a stepwise trend of increasing temperatures accompanied by short-term cooling events is documented from megafloras for the Early Miocene in Central Europe (Early Aquitanian; Mosbrugger et al., 2005). The MCO reflects the warmest episode of the Neogene with minimum global ice volume (Holbourn et al., 2007, 2014, 2015; Jiménez-Moreno and Suc, 2007; Mudelsee et al., 2014; Steinthorsdottir et al., 2021b). The drivers of MCO warming and its geographic and temporal variability during the MCO are still debated. The onset of the MCO coincides with the 17–16 Ma Columbia River Basalt eruptions in western North America (Longman et al., 2022; Steinthorsdottir et al., 2021b). Combined with increased fluxes of $CO_2$ into the atmosphere caused by volcanic arc or rift activity and decreased continental weatherability, the Columbia River Basalt eruptions may have triggered the transition into the warm MCO (Longman et al., 2022 and ref. therein). Although the MCO was generally an interval of intense warmth, high-resolution, benthic




foraminiferal $\delta^{18}$O records also indicate high-amplitude climate variability, particularly towards its end (15.7–14.7 Ma; Holbourn et al., 2007, 2014, 2022).

Whereas the mid-latitude N-S temperature gradient in Western Europe was probably reduced during the Middle Miocene (Jiménez-Moreno and Suc, 2007), continental paleoclimate reconstructions of the MCO emphasize a strong semi-arid-to-humid latitudinal gradient with annual precipitation amounts lower than today (i.e., herpetological data from Böhme, 2004; mammal diversity from Costeur and Legendre, 2008; pollen records from Jiménez-Moreno and Suc, 2007). Southern mid-latitude locations, such as south-central Spain, likely experienced semi-arid climatic conditions with open environments (subdesertic conditions; Urban et al., 2010), whereas more

northern regions, e.g., central eastern France and Switzerland, were characterized by a more humid climate (1000-1500 mm/yr; Böhme, 2004) and pronounced precipitation seasonality (Botsyun et al., 2022; Costeur and Legendre, 2008 and ref. therein). The existence of a Middle Miocene latitudinal aridity gradient stretching from southern to northern mid-latitudinal Europe may have resulted from a trade wind system, which was characterized by prevalent northeasterly winds. Orography may have contributed to the zonal precipitation pattern: trade winds

produced summer precipitation upwind and dryness downwind of the east-west-oriented European orogens (Pyrenees and Alps; Böhme, 2004). The demarcation between westerlies and trade winds likely positioned north of 47ºN paleolatitude (Böhme, 2004). Paleoclimate modelling suggests this pattern had ceased by the Late Miocene with a shift of European atmospheric circulation towards westerly-dominated winds (Quan et al., 2014).

After the MCO, significant cooling led to the renewed expansion of the Antarctic ice-sheet and deep-water cooling

during the MMCT (14.7–13.8 Ma; Holbourn et al., 2014; Miller et al., 1991). The latter had a profound impact on marine and terrestrial realms, e.g., the global reorganization of ocean circulation patterns (Holbourn et al., 2022), the increase of foraminifera taxa, the decline in crustacean diversity (Steinthorsdottir et al., 2021b and ref. therein), the gradual disappearance of subdesertic plants in southwestern Europe (Jiménez-Moreno and Suc, 2007), and the extinction of Central European alligators and giant turtles (Böhme, 2004). The MMCT also impacted terrestrial

temperature and precipitation patterns with enhanced temperature seasonality and a shift towards a westerly-dominated wind regime over Central Europe by the Tortonian (Methner et al., 2020; Quan et al., 2014).

### 2.3 Present-day Mediterranean climate

The main characteristics of the present-day climate in the circum-Mediterranean region, a marked seasonality between mild, maritime, wet winters and warm to hot, dry summers (Lionello, 2012; Seager et al., 2019) have

been recorded as early as the Pliocene (Suc, 1984). The timing of the onset of the characteristic marked seasonality is however still unclear. Typically, North Atlantic storm tracks deliver precipitation to the region during the winter months. Below-freezing temperatures rarely occur during these months, except in mountainous areas. During the summer season, mid-latitude storm tracks weaken due to the dominance of the Azores High pressure system, resulting in warm to hot and dry summers (Seager et al., 2019; Ulbrich et al., 2012).

Being influenced by both, a mid-latitude climate variability steered by the North Atlantic Oscillation (NAO) and subtropical conditions due to the descending branch of the Hadley cell, the overall temperature and precipitation



patterns in the circum-Mediterranean region reflect the interaction of many individual drivers, including orography, latitude, and land-sea distribution (Ulbrich et al., 2012).

**3. Methodology**

**3.1. Pedogenic carbonate horizons and sampling strategy**

Stratigraphic correlations between the investigated sections permit the construction of a ca. 500 m long composite section (Châteauredon composite section; BCR) that consists of three correlated subsections: Beynes (BE), Ruines (RU) and Les Granges (GR; Fig. 1C; Bialkowski et al., 2006; Cojan et al., 2013; Gillot et al., 2022). Based on
bio- and $\delta^{13}$C-chemostratigraphy, the BCR covers the Early to Middle Miocene from ca. 23 to ca. 13 Ma (Bialkowski et al., 2006; Cojan et al., 2013). Due to its proximity to the Miocene shoreline, the stratigraphy of the BCR consists of alluvial, lacustrine, fluvial, marine, and coastal facies with intercalations of continental deposits including numerous paleosol horizons. More than 90% of the paleosol horizons can be classified as calcisols with irregularly shaped and globular, hard micritic pedogenic carbonate nodules dispersed in a fine-grained mudstone,
shale, or siltstone matrix (Cojan and Gillot, 2022). The pedogenic carbonate nodules formed on floodplain alluvium and are found in isolated accumulation (Bk) horizons. The nodules range from a few millimeters to several centimeters in diameter and are commonly spheroidal with irregularities (see Fig. 2C).

In this study, sixteen pedogenic carbonate samples from different Bk horizons within the BCR were analyzed for $\delta^{13}$C, $\delta^{18}$O, and $\Delta_{47}$-paleothermometry. Individual pedogenic carbonate nodules are from Bk horizons of
carbonate-rich mature paleosols and correspond to the sampling horizons of Bialkowski (2002) and complement the existing $\delta^{13}$C record (Bialkowski et al., 2006; Figs. 2, S1). Nodules were sampled at >50 cm depth (Bialkowski, 2002). Two pedogenic carbonate nodules come from the fine-grained alluvial succession of the Beynes section, which is dominated by pink to reddish marls of the Molasse Rouge. Fourteen pedogenic carbonate nodules originate from the transgression-regression cycles of the Molasse Grise and the Marine Formation of the Ruines
and Granges sections (Table 1, Fig. S3).





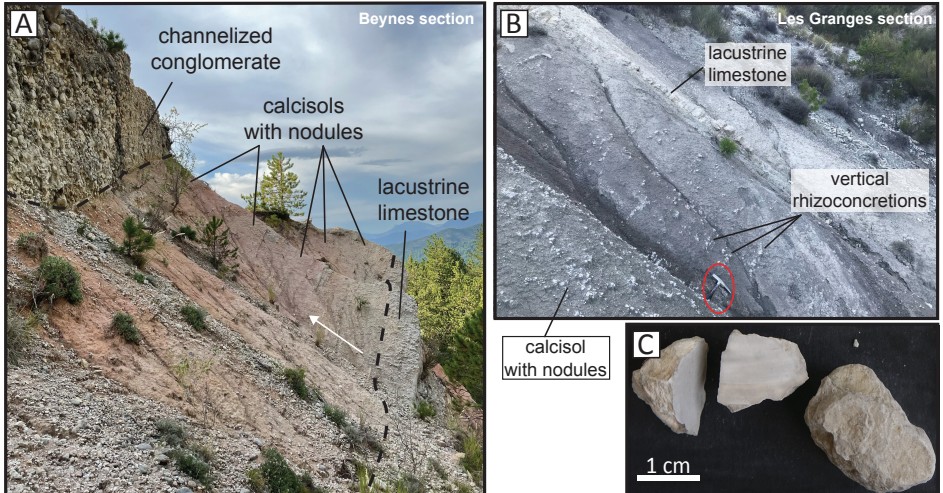

**Figure 2:** Field photographs. (A) Beynes Section: paleosols that developed in distal alluvial fan deposits (channelized conglomerates; modified after Cojan and Gillot, 2022). White arrow indicates younging direction of stratigraphy orientation. (B) Les Granges Section: stacked horizons of pedogenic carbonate-rich paleosols that contain vertical rhizoconcretions and are overlain by lacustrine limestone. Hammer for scale. (C) Pedogenic carbonate nodules from the Les Granges Section.

### 3.2. Age model

The initial age model for the BCR composite section relies on integrated biostratigraphy and chemostratigraphy ($\delta^{13}$C values of pedogenic carbonate; Bialkowski et al., 2006; Cojan et al., 2013). The age model is based on the correlation of trends in pedogenic carbonate $\delta^{13}$C values (n= 76) with trends in marine reference $\delta^{13}$C records (compilation of ODP-DSDP sites, Bialkowski et al., 2006 and ref. therein). Each correlated time interval of the BCR section was validated by biostratigraphy (micromammal faunas and gastropod fossils) and palynological analysis (dinocysts, spores, and pollen, including a distinct *Avicennia* peak; Châteauneuf et al., 2006). The initial age model adopted the time scale of Berggren et al. (1995).

Here, we adopt and update the initial age model by tying the $\delta^{13}$C curve of the BCR composite section to the more recent benthic deep-sea foraminifera carbon isotope curve from Westerhold et al. (2020) by using the *QAnalySeries* software (Kotov and Paelike, 2018). We use the same tie points (e.g., positive or negative $\delta^{13}$C excursions) between the continental and reference marine records as Bialkowski et al. (2006) and Cojan et al. (2013). For intervals with limited tie points and matching trends, the correlation was based on the best fit provided by the software assuming uniform sedimentation rates and ensuring similar trends with the reference marine $\delta^{13}$C curve (Fig. S4; Tables S2 and S3).

Age uncertainties in the DVB range from 100 ka to 1 Ma (Fig. 3; Bialkowski, 2002). The age uncertainties in the DVB are constrained by the comparison of the DVB biostratigraphy with the biostratigraphy of the well-correlated magnetostratigraphic NAFB and the well-described biostratigraphy of the Aquitaine basin (Bialkowski, 2002).



### 3.3. Carbonate stable isotope analysis and $\Delta_{47}$ thermometry

Clumped isotope thermometry ($\Delta_{47}$) relies on the determination of the temperature-dependent excess abundance of rare carbon and oxygen isotope bonds ($^{13}$C-$^{18}$O), the so-called clumping, within the carbonate crystal lattice (Eiler, 2011). The isotopic clumping of carbonate molecules is dependent on the temperature of carbonate formation, but independent of the $\delta^{18}$O value of the fluid from which the carbonate precipitated (Bernasconi et al., 2018; Eiler, 2011).

Isotope analyses were performed at the Joint Goethe University-Senckenberg BiK-F Stable Isotope Facility (Frankfurt, Germany). Measurements were performed on a Kiel IV Carbonate Device coupled to a MAT 253plus IRMS. For a detailed method description, we refer to the Supplementary Material. Prior to measurements, carbonate nodules were cut and subsequently powdered with a dental drill.

Samples were analyzed in a variable number of replicates (n = 8-18) using between 90 and 190 μg of material
according to carbonate content. Samples were analyzed together with the ETH-1 – ETH-4 carbonate standards (Bernasconi et al., 2018). Raw intensity data were corrected for the negative pressure baseline using the m/z$_{47.5}$ signal and scaling factor method (Bernecker et al., 2023; Fiebig et al., 2021) applied to ETH-1 and ETH-2. Standardization was carried out by utilizing the D47crunch module (Daëron, 2021) in pooled sessions mode with ETH standards as carbonate anchors ($\Delta_{47\ \text{I-CDES}}$, Bernasconi et al., 2021). The universal temperature calibration of
Anderson et al. (2021) is used for temperature conversion.

Stable isotope results of pedogenic carbonates are reported in per mil deviation relative to V-PDB ($\delta^{13}$C$_{\text{carb}}$, $\delta^{18}$O$_{\text{carb}}$). $\delta^{18}$O$_{\text{w}}$ was calculated based on our obtained carbonate formation temperature (T($\Delta_{47}$)) and $\delta^{18}$O$_{\text{carb}}$ data and the oxygen isotope equilibrium fractionation equation of Kim et al. (2007). $\delta^{13}$C$_{\text{carb}}$ and $\delta^{18}$O$_{\text{carb}}$ errors are reported as 1 standard deviation (±SD), 2 standard error (2 SE), and 1 standard error (1 SE; Tables 1 and S1).

### 4. Results

Complete results of stable and clumped isotope measurements of the pedogenic carbonate nodules are presented in Table S1. The analyses of pedogenic carbonate nodules of the same horizon (see Methodology) complement data presented in (Bialkowski, 2002) (Table 1). The following sections discuss the results from analyses carried out in this study.

### 4.1. Stable ($\delta^{18}$O$_{\textbf{carb}}$ and $\delta^{13}$C$_{\textbf{carb}}$) and clumped ($\Delta_{47}$) isotope data

$\delta^{18}$O$_{\text{carb}}$ values of paleosol carbonate samples from the Digne-Valensole basin range from -7.02 ‰ to -4.66 ‰ (V-PDB), and $\delta^{13}$C$_{\text{carb}}$ values range from -9.90 ‰ to -6.15 ‰ (V-PDB). The $\Delta_{47\ \text{(I-CDES)}}$ values of pedogenic carbonate range from 0.565 ‰ to 0.619 ‰, which translates to a temperature range of 35.2 ºC to 16.8 ºC. External standard errors (1 SE) for 8 to 18 replicate measurements range from 0.01 ‰ to 0.02 ‰, which corresponds to temperature
uncertainties covering 3.5 ºC to 6.0 ºC. Calculated soil water $\delta^{18}$O$_{\text{w}}$ values based on corresponding $\delta^{18}$O$_{\text{carb}}$ and $\Delta_{47}$ values range from -6.29 ‰ to -0.51 ‰ (V-SMOW).

Due to potential diagenetic alteration, measurements from one sample (98GR22) are not included in the interpretation of the results and discussion (Table S1, Fig. S2).



### 4.2. Paleosol temperature reconstruction

We report stable and clumped isotope data of our DVB record within four intervals, which closely follow Miocene stages (Fig. 3). Interval I (Aquitanian, 22.3–21.1 Ma) yields relatively similar T($\Delta_{47}$) values with an average temperature of 27.6 ºC (n = 2; SD = 1.6 ºC), while data from interval II (Burdigalian, 19.6–16.7 Ma) show positive trends in $\delta^{13}C_{carb}$ (from ca. -7.4 ‰ to -6.9 ‰) and $\delta^{18}O_{carb}$ values (from ca. -6.6 ‰ to -6.0 ‰) and an average $\Delta_{47}$ temperature of 32.0 ºC (n = 4; SD = 3.5 ºC; Fig. S3). Interval III (Langhian–Early Serravallian, 16.0–13.8 Ma) is

characterized by high-amplitude and rapid temperature variations ranging from 20.6 ºC to 34.7 ºC (n = 7; SD = 5.3 ºC; Fig. 3) with corresponding changes in the $\delta^{18}O_w$ record. Overall, $\delta^{13}C_{carb}$ and $\delta^{18}O_{carb}$ values follow a positive trend from 18.6 Ma to 14.5 Ma (Fig. S3).

Interval IV (Serravallian, 13.4–13.2 Ma) is characterized by the two lowest $\Delta_{47}$ temperatures in our record of 20.5 ºC and 16.8 ºC (13.4 and 13.2 Ma, respectively; average 18.6 ºC; n = 2; SD = 2.6 ºC).

In conclusion, we observe the following patterns in our Miocene record:

The Aquitanian data reflects similar temperatures averaging 27.6 ºC. During the Burdigalian and the Langhian, we observe three key elements in our data: a) $\Delta_{47}$ temperatures decrease from ca. 33 ºC (18.3 Ma) to 20 ºC (15.0 Ma); b) $\Delta_{47}$ temperatures increase to ca. 34 ºC at the end of the Langhian and start of the Serravallian; c) $\Delta_{47}$ temperatures decrease at 13.8 Ma to the lowest measured $\Delta_{47}$ temperatures. The rapid large magnitude decrease in

$\Delta_{47}$ temperature from 34.3 ºC to 16.8 ºC between 13.8 and 13.2 Ma is accompanied by a 1.5 ‰ decrease of $\delta^{18}O_{carb}$ values from -5.5 ‰ to -7.0 ‰, which corresponds to a decrease in $\delta^{18}O_w$ values of 5.1 ‰ from -1.2 ‰ to -6.3 ‰.

**Table 1.** Soil carbonate $\delta^{18}O_{carb}$, $\delta^{13}C_{carb}$ and T($\Delta_{47}$) values, and calculated oxygen isotopic ratios of soil water $\delta^{18}O_w$.

| Interval | Sample ID | Section | Age (Ma) | Age err (Ma) | n | $\Delta_{47-CDES}$ (‰) | $\sigma$ (‰) | ± 1 SE (‰) | T($\Delta_{47}$) (°C) | - 2 SE (°C) | + 2 SE (°C) | $\delta^{13}C_{carb}$ (‰, vPDB) | $\delta^{18}O_{carb}$ (‰, vPDB) | $\delta^{18}O_w$ (‰, vSMOW) |
|---|---|---|---|---|---|---|---|---|---|---|---|---|---|---|
| IV | 99CH03 | GR | 13.18 | +0.1, -0.5 | 16 | 0.619 | 0.0376 | 0.0118 | 16.76 | 6.94 | 7.48 | -9.9 | -7.02 | -6.29 |
| IV | 99CH01 | GR | 13.37 | 0.1 | 13 | 0.607 | 0.0826 | 0.0135 | 20.51 | 8.22 | 8.98 | -8.66 | -6.61 | -5.07 |
| III | 99CH16 | GR | 13.78 | 0.1 | 14 | 0.568 | 0.0335 | 0.0123 | 34.28 | 8.62 | 9.41 | -6.15 | -5.55 | -1.23 |
| III | 99CH14 | GR | 13.87 | 0.1 | 11 | 0.571 | 0.0549 | 0.0141 | 33.24 | 9.67 | 10.69 | -8.24 | -4.66 | -0.51 |
| III | 98GR34 | GR | 14.23 | 0.1 | 17 | 0.567 | 0.0438 | 0.0111 | 34.69 | 7.79 | 8.43 | -6.24 | -6.13 | -1.75 |
| III | 99CH10 | GR | 14.61 | 0.1 | 17 | 0.589 | 0.0231 | 0.0113 | 26.83 | 7.35 | 7.93 | -6.28 | -5.78 | -2.92 |
| III | 98GR29 | GR | 14.95 | 0.1 | 17 | 0.607 | 0.0421 | 0.0113 | 20.61 | 6.94 | 7.47 | -6.56 | -5.87 | -4.29 |
| III | 98GR24 | GR | 15.58 | 0.1 | 11 | 0.588 | 0.042 | 0.0141 | 26.87 | 9.14 | 10.04 | -6.9 | -5.78 | -2.91 |
| III | 98GR18 | GR | 16.01 | 0.1 | 18 | 0.591 | 0.0467 | 0.0108 | 25.83 | 3.76 | 6.99 | -6.74 | -5.74 | -3.08 |
| II | 98GR08 | GR | 16.67 | 0.15 | 18 | 0.565 | 0.0192 | 0.0108 | 35.21 | 4.11 | 7.62 | -6.86 | -5.95 | -1.47 |
| II | 98GR02 | GR | 17.29 | 0.15 | 15 | 0.577 | 0.052 | 0.012 | 30.77 | 4.41 | 8.1 | -7.14 | -6.08 | -2.45 |



| | | | | | | mean $\Delta_{47\text{-CDES}}$ (‰) | mean $\sigma$ (‰) | mean ± SE (‰) | mean T($\Delta_{47}$) (°C) | mean - 1 SE (°C) | mean + 1 SE (°C) | mean $\delta^{13}C_{carb}$ (‰, vPDB) | mean $\delta^{18}O_{carb}$ (‰, vPDB) | mean $\delta^{18}O_w$ (‰, vSMOW) |
|---|---|---|---|---|---|---|---|---|---|---|---|---|---|---|
| II | 98RU32 | RU | 18.32 | 1.0 | 14 | 0.568 | 0.0418 | 0.0125 | 34.35 | 4.75 | 8.69 | -7.32 | -6.93 | -2.64 |
| II | 98RU30 | RU | 18.63 | 1.0 | 10 | 0.587 | 0.0275 | 0.0147 | 27.49 | 5.27 | 9.54 | -7.38 | -6.60 | -3.63 |
| I | 98RE46 | BE | 21.14 | 0.15 | 8 | 0.583 | 0.0341 | 0.0163 | 28.78 | 5.97 | 10.67 | -7.74 | -6.41 | -3.19 |
| I | 98RE30 | BE | 22.33 | 0.15 | 10 | 0.590 | 0.0387 | 0.0148 | 26.49 | 5.25 | 9.51 | -7.48 | -6.75 | -3.99 |
| Interval IV | | 13.37–13.18 | | | 2 | 0.613 | 0.060 | 0.013 | 18.63 | 3.79 | 4.11 | -9.28 | -6.81 | -5.67 |
| Interval III | | 16.01–13.78 | | | 7 | 0.583 | 0.041 | 0.012 | 28.91 | 4.04 | 4.39 | -6.73 | -5.65 | -2.38 |
| Interval II | | 19.63–16.67 | | | 4 | 0.574 | 0.035 | 0.013 | 31.96 | 4.24 | 4.64 | -7.18 | -6.39 | -2.55 |
| Interval I | | 22.33–21.14 | | | 2 | 0.586 | 0.036 | 0.016 | 27.64 | 5.05 | 5.61 | -7.61 | -6.582 | -3.59 |

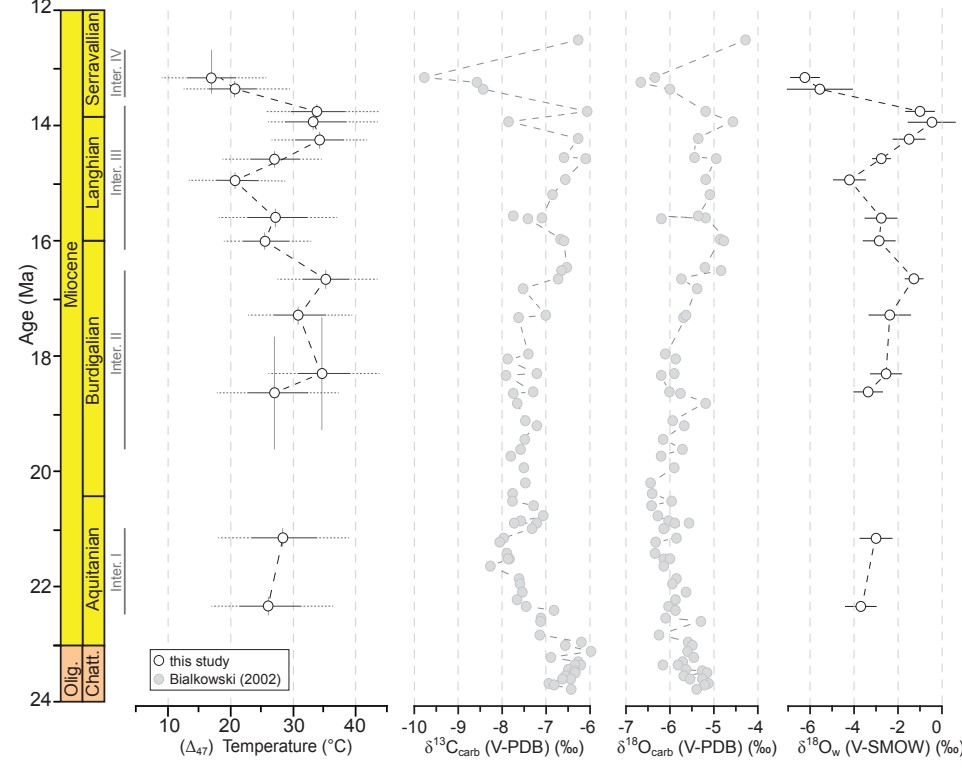

**Figure 3:** Soil carbonate $\delta^{18}O$, $\delta^{13}C$ (Bialkowski, 2002) and T($\Delta_{47}$) (this study) values. Oxygen isotopic ratios of soil water ($\delta^{18}O_w$) were calculated with $\delta^{18}O_{carb}$ measured in this study; timing of Intervals I–IV is indicated.




**5. Discussion**

**5.1 Pristine stable isotope values**

We consider the obtained $\Delta_{47}$-temperatures pristine and to reflect pedogenic carbonate growth temperatures for the following reasons: (1) our $\Delta_{47}$-temperatures align with the range of temperatures found in the upper layers of Earth's subsurface, which are mainly controlled by MAT (Molnar, 2022; Mosbrugger et al., 2005); (2) within the reported samples there is no evidence of (localized) fluid flow and associated recrystallization (Quade et al., 2013 and ref. therein), which is attested by thin sections of nodules that reveal a dominantly micritic texture (Fig. S1);

(3) the $\delta^{18}O_{carb}$ values fall into the range attributed to unaltered pedogenic carbonates (Tabor and Myers, 2015); (4) the burial depth of the sampled paleosol sequences in the DVB was relatively shallow (< 500 m; Ford et al., 1999) and there is no overall trend to higher temperatures within the more deeply buried parts of the section.

However, one pedogenic carbonate sample (99GR22) contains sparry calcite veins. This sample may have been affected by diagenetic alteration and its relatively high $\Delta_{47}$ temperature (39.2 °C; Table S1; Fig. S2) may not

reflect pedogenic carbonate formation temperatures but rather carbonate formation during diagenesis. Consequently, the sample was not included in the discussion.

**5.2 Seasonality of soil carbonate precipitation**

We observe large (up to 17.5 °C) variations in T($\Delta_{47}$) soil carbonate formation temperatures in the DVB record. Such large temperature variations cannot be explained by regional or global terrestrial temperature change alone

(see e.g., Fauquette et al., 2015; Mosbrugger et al., 2005; Westerhold et al., 2020).

In situ pedogenic carbonate formation occurs within soil profiles in the presence of soil water in semi-arid to sub-humid climates. Pedogenic carbonates most commonly form during the warm and dry season when plants are active and evapotranspiration attains a maximum (Quade et al., 2007, 2013; Zamanian et al., 2016). As such, carbonate precipitation responds to interannual changes in soil moisture availability and is mainly controlled by

the interplay between soil water availability, soil temperature, and soil $CO_2$ levels in the presence of sufficient $Ca^{2+}$. Net calcite accumulation occurs when (re-)precipitation of calcite exceeds calcite dissolution during seasonal wet and dry cycling (Zamanian et al., 2016). This implies that changes in the timing and/or duration of the pedogenic carbonate growth season will affect $\Delta_{47}$ temperatures. Because soil drying is an important requirement for pedogenic carbonate formation, a change in the timing of the dry season would inevitably result in a shift of

the pedogenic carbonate formation period (Fig. 4). Consequently, recorded $\Delta_{47}$-temperatures may vary depending on the timing of rainfall seasonality and hence seasonality of carbonate formation (Fig. 4A-B; Gallagher et al., 2019; Kelson et al., 2020). In accordance with other studies of pedogenic carbonate T($\Delta_{47}$) from Central Europe (e.g., Methner et al., 2020) we suggest that a seasonal change in the timing of annual carbonate precipitation may have played an important role in controlling the observed temperature pattern, i.e., a change of pedogenic

carbonate precipitation from e.g., the cool(er) to the warm(er) season or vice versa. Presently, summer temperatures average ca. 19 °C and winter season temperatures average ca. 2 °C in the DVB (IAEA/WMO, 2024). Soil temperatures are typically close to air temperatures throughout the year and the variation in air and soil temperatures is likely similar year-round (Molnar, 2022; Sheldon and Tabor, 2009). A hypothetical, instantaneous



shift from warm to cold season carbonate precipitation or vice versa would therefore presently account for a

change of up to 17 °C in soil carbonate formation temperatures.

Based on the above, we suggest that changes in the $\Delta_{47}$ soil carbonate formation temperatures throughout the Middle Miocene are best explained by warming or cooling during the MCO and MMCT, respectively, in combination with modifications of the intra-annual timing of soil carbonate formation.

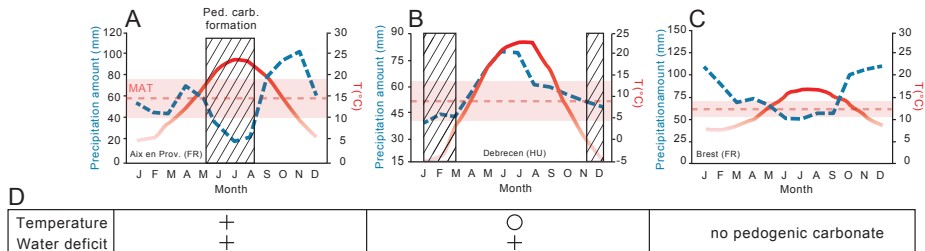

**Figure 4:** Variations in seasonality of precipitation may drive changes in pedogenic carbonate formation temperatures. Depending on the seasonality of pedogenic carbonate formation, soil temperatures may reflect (A) warm (> MAT) or (B) cool (≤ MAT) temperatures. (C) Conditions for the formation of pedogenic carbonate are not met because of the lack of contrasting seasonality and temperature throughout the year. Each scenario corresponds to present-day climates at different locations in Western and Central Europe. (D) Influence (cross: large, circle: moderate) of the interplay between soil temperature and soil

water deficit on the formation of pedogenic carbonate. Temperature dominated (A), water deficit dominated (B), and no pedogenic carbonate formation (C) scenarios.

A change in seasonality of carbonate precipitation explaining the large observed temperature variations in our $\Delta_{47}$ record is supported by the large variations in calculated $\delta^{18}O_w$ during the MCO and MMCT of up to 3.8 ‰ and

5.0 ‰, respectively. These $\delta^{18}O_w$ variations may reflect changes in atmospheric circulation patterns and/or a change in the season of moisture delivery, which would both be associated with different moisture sources. Consequently, a change in the seasonal timing of precipitation in the DVB is likely to be linked with a change in moisture source(s) over the region.

A shift in precipitation seasonality over Western (this study), as well as Central Europe (Methner et al., 2020) throughout the Middle Miocene requires a profound reorganization of atmospheric circulation. The distribution of fossil snakehead fish at the start of the MCO suggests moist summer conditions which may be reached by a poleward shift of the Hadley circulation (Böhme, 2004). This may have resulted in a southeastward transport of air masses over the North Sea Basin, the Paratethys, and along the Alpine chain transporting moist air over Central

Europe during the summer months (Böhme, 2004) which is supported by tree fall directions and the eccentricity of tree growth rings (Utescher et al., 2000). After the MCO, a permanent shift from a northeasterly to a westerly-dominated wind regime (similar to the present-day Mediterranean-type climate) over the northern Mediterranean region is suggested to have occurred at the latest before the Tortonian (ca. 12 Ma; Quan et al., 2014).



The data presented here do not permit to identify drivers and mechanisms of changes in atmospheric circulation, but we consider it likely that an alternation between northeasterly winds and westerlies or latitudinal variability in the main westerlies track throughout the MCO contributed to modifying the pattern of wet and dry seasons and hence the temperature fluctuations observed in our $\Delta_{47}$ soil carbonate data.

We suggest that the DVB MCO and MMCT record reveals two distinctly different climatic patterns. During the Langhian, the climate is marked by a dry cool season. Soil drying and pedogenic carbonate formation during late fall and/or early winter (Fig. 4B) is reflected in relatively low $T(\Delta_{47})$ and $\delta^{18}O_w$ values. In this 'dry winter' scenario, rainfall occurred mostly during warmer (summer) months and was possibly delivered by moist air masses from the North Atlantic Ocean. We hypothesize that a transition to a climate similar to the present-day

Mediterranean-type climate ('proto-Mediterranean climate'), characterized by hot and dry summers (Fig. 4A), occurred during the Middle Miocene. This implies that already during certain time intervals of the MCO a climate similar to the present-day Mediterranean climate existed. In such a climate state, the drying of soils during the summer season led to dominant pedogenic carbonate formation in early summer, with associated high $T(\Delta_{47})$ and $\delta^{18}O_w$ values. Finally, our data are consistent with a return to the 'dry winter' scenario in the early Serravallian

(13.4–13.2 Ma) resulting in relatively low $\Delta_{47}$-temperatures and $\delta^{18}O_w$ values. We therefore propose that high $T(\Delta_{47})$ (> 30 ºC) and high $\delta^{18}O_w$ values (> -2 ‰), during the MCO and MMCT resulted from a pronounced Mediterranean moisture source, whereas lower $\Delta_{47}$-temperatures and lower $\delta^{18}O_w$ values (< -2 ‰) reflect predominance of a North Atlantic moisture source.

### 5.3 Middle Miocene northern Mediterranean and central European terrestrial paleoclimate

Irrespective of the change in the seasonality of the soil carbonate formation, three main elements characterize our paleosol temperature data and include: (1) the transition into the MCO (17.3–16.7 Ma) during the Burdigalian, which is reflected by an increase in $\Delta_{47}$-temperatures and the highest temperature of the record (35.2 ºC), (2) overall cooling from the onset to the end of the MCO, (3) a large temperature increase during the MMCT, and (4) a distinct (> 15 ºC) and rapid (ca. 600 ka) cooling phase accompanied by a significant decrease in $\delta^{18}O_w$ (-5.0 ‰)

after ca. 13.8 Ma, which coincides with global post-MMCT cooling.

     In the following, we compare the DVB record with time-equivalent terrestrial data from the northern Mediterranean region and Central Europe (Fig. 5).

     During the Aquitanian (22.3–21.1 Ma) and Middle Burdigalian (18.6–17.3 Ma), $T(\Delta_{47})$ values of the DVB region

are significantly (up to 10 ºC) higher than estimated mean annual temperatures (MAT) for the DVB derived from pollen (Fauquette et al., 2015) and paleosol chemistry (Gillot et al., 2022). During the same time interval, our $T(\Delta_{47})$ are also up to 10 °C higher than MAT reconstructed from Central European megafloras (fruits, seeds, and leaves), but in good agreement with estimated warm month temperatures (WMT) (Mosbrugger et al., 2005; Fig. 5).



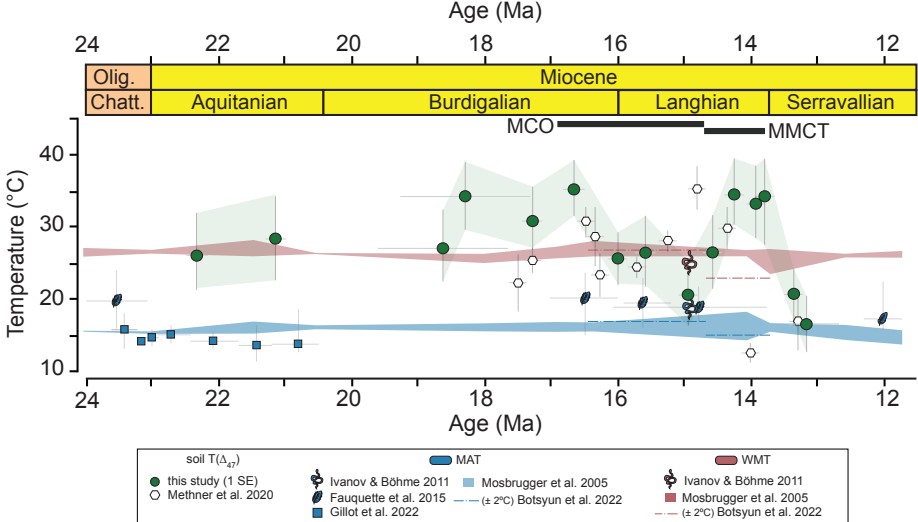

**Figure 5:** Reconstructed Miocene mid-latitude terrestrial temperatures from Western and Central Europe. Temperature reconstructions based on carbonate clumped isotope thermometry from the Digne-Valensole basin are compared to T($\Delta_{47}$) data from the NAFB (Methner et al., 2020), mid-latitude European mean annual temperatures (MAT) and warm month temperatures (WMT) inferred from terrestrial (paleofloral; Fauquette et al., 2015; Mosbrugger et al., 2005; herpetofauna and climofunction from paleosols; Ivanov and Böhme, 2011; Gillot et al., 2022) data, as well as modeled MAT and WMT values (Botsyun et al., 2022) for the paleogeographic position of the DVB.

At the onset of the MCO, our DVB T($\Delta_{47}$) record shows a warming trend similar to that observed in the T($\Delta_{47}$) record from the Swiss NAFB paleosols (Methner et al., 2020).

At ca. 15 Ma, we observe colder $\Delta_{47}$-temperature that aligns particularly well with MAT values derived from herpetological data (Ivanov and Böhme, 2011), pollen records (Fauquette et al., 2015), and climate models (Botsyun et al., 2022). This T($\Delta_{47}$) value in the DVB suggests a switch from a dry season, summer bias towards conditions indicating a 'dry winter' scenario with late fall/early winter soil carbonate formation (see section 5.2). Conversely, during the subsequent warming trend leading into the onset of the MMCT (ca. 15.5–14.0 Ma), we observe T($\Delta_{47}$) values that converge towards temperatures above the paleobotanical WMT values, and hence a return to summer dry, warm-season biased temperatures .

Finally, we attribute the large magnitude temperature decrease after ca. 14 Ma, to post-MMCT continental cooling. The climate change after 14 Ma is expressed in an impoverishment in plant diversity in Western Europe (Jiménez-Moreno and Suc, 2007) and led to the regional extinction of temperature-sensitive tetrapods in Central Europe, such as alligators, giant turtles, and chameleons (Ivanov and Böhme, 2011). The temperatures in the DVB at 13.5 Ma, following the post-MMCT cooling, are equivalent to the NAFB $\Delta_{47}$ soil temperatures and herpetological, botanical and modeled-based MAT (Fig. 5; Botsyun et al., 2022; Fauquette et al., 2015; Ivanov and Böhme, 2011; Methner et al., 2020). We can only attempt to decipher the contributions of seasonal temperature change and overall regional cooling to the ca. 17.5 °C decrease in soil carbonate formation temperatures visible in the DVB

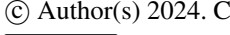



$\Delta_{47}$-record. Assuming that the temperature change of ca. 14 ºC (20.6 ºC to 34.7 ºC) that is observed between 14.9 Ma and 14.2 Ma reflects only the change in carbonate formation season without any major regional temperature change, we posit that up to $14 \pm 2$ ºC of the 17.5 °C temperature decrease between 13.8 and 13.2 Ma is attributable to a seasonality shift in the DVB. We therefore propose that the remaining 3-4 °C temperature decrease after 13.8 Ma reflects the effect of regional post-MMCT cooling. This value is comparable to the drop in the minimal

summer temperature of ca. 14 ºC between 14.0 and 13.5 Ma described from herpetological data for Central Europe (Böhme, 2003).

Therefore, the significant decrease (17.5 ºC) in $\Delta_{47}$-temperatures and associated decrease in $\delta^{13}C_{carb}$, $\delta^{18}O_{carb}$, and calculated $\delta^{18}O_w$ values at or after 13.9 Ma supports the interpretation that regional cooling superimposed by a component of changing climate seasonality are accountable for the decrease in T($\Delta_{47}$) (Methner et al., 2020).

**5.4 Continental soil temperatures related to marine records**

Throughout the MCO and MMCT, our northern Mediterranean terrestrial soil $\Delta_{47}$-temperatures show significant similarities with marine records (Fig. 6). The MCO is characterized as a period of sustained global warmth marked by high-amplitude climate variability (Fig. 6A) (Holbourn et al., 2007, 2014, 2015; Steinthorsdottir et al., 2021b). Our terrestrial record agrees with global records and indicates overall high temperatures with pronounced

temperature variability throughout the MCO.

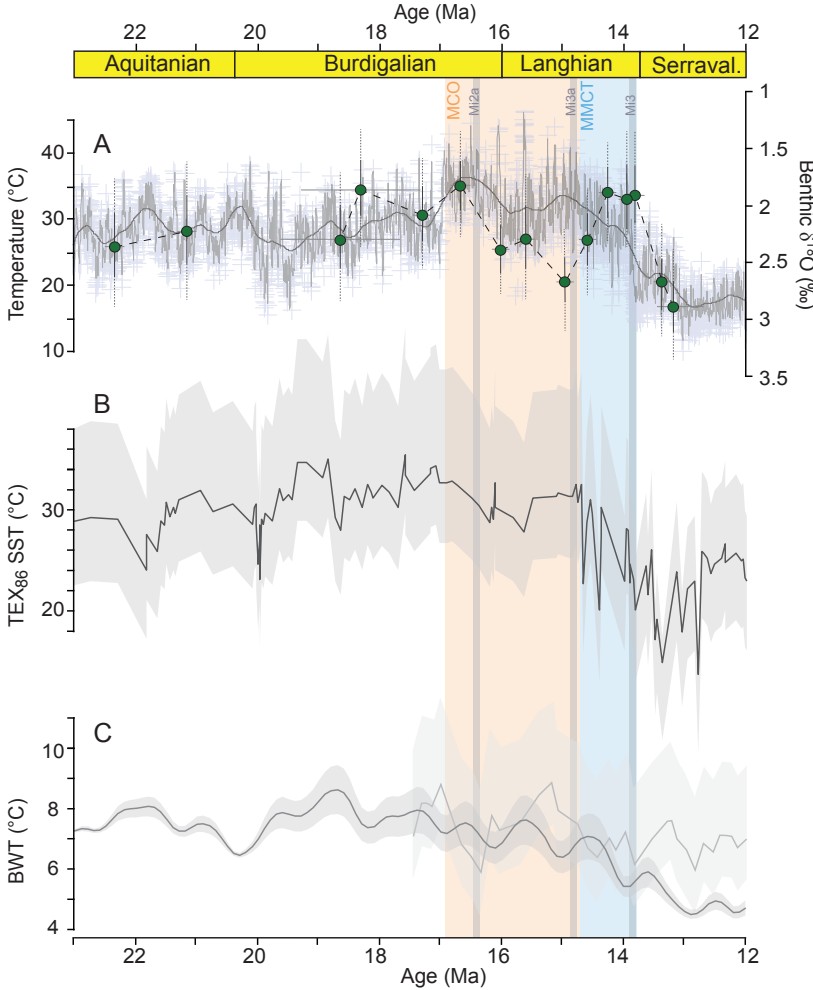

**Figure 6:** Clumped isotope-based reconstruction of terrestrial temperatures compared with global marine data, within the timing of the MCO and MMCT (Westerhold et al., 2020), and Mi-events (Miller et al., 2020). (A) $\Delta_{47}$-temperatures from the Digne-Valensole basin (in green) and benthic oxygen isotopic curve (Westerhold et al., 2020). (B) TEX$_{86}$-based sea surface temperatures of ODP site 608, North Atlantic (Super et al., 2018). (C) Bottom-water temperatures from Cramer et al. (2011) in dark grey and Lear et al. (2015) in light grey.

Following a period of rapid warming and/or polar ice melting at 16.9 Ma at the onset of the MCO, the Middle Miocene was globally marked by fundamental changes in the global carbon reservoir resulting in $\delta^{18}O_{benthic}$ and $\delta^{13}C_{benthic}$ increases (Fig. 6A) (Holbourn et al., 2015, 2022; Westerhold et al., 2020) and cooling in sea surface (Fig. 6B) and bottom water temperature (BWT; Fig. 6C) from 16.5 to ca. 16.0 Ma (Cramer et al., 2011; Lear et



al., 2015; Super et al., 2018). Those results align well with the cooling phase recorded in our T($\Delta_{47}$) between circa 16.7 and 15.6 Ma.

Within age uncertainty, our coolest $\Delta_{47}$-temperature within the MCO coincides with $\delta^{18}O_{benthic}$ excursion Mi3a, which is associated with significant changes in global temperature and ice-volume history (Fig. 6A,C; Holbourn et al., 2007; Miller et al., 2020).

The subsequent T($\Delta_{47}$) warming trend into the MMCT from 14.9 Ma until 14.2 Ma in our continental record is also observed in benthic records (Fig. 6A,C; Cramer et al., 2011; Westerhold et al., 2020). Finally, our rapid post-MMCT cooling (after ca. 13.8 Ma) is associated with a major decrease in $\delta^{13}C_{carb}$ (-4.0 %), $\delta^{18}O_{carb}$ (-2.5 %), and $\delta^{18}O_w$ (-5.0 %) values and parallels the stabilization of the East Antarctic Ice Sheet, expressed in benthic $\delta^{18}O$ excursion Mi3, and associated sea level decline, and a decrease of BWT (Holbourn et al., 2015; Miller et al., 2020; Sosdian et al., 2020; Steinthorsdottir et al., 2021b; Westerhold et al., 2020). It coincides also particularly well in time with cooling in continuous marine successions from the Eastern Mediterranean region at 13.82±0.03 Ma (Abels et al., 2005).

Results of our study documenting the paleosol $\Delta_{47}$-temperatures of the Digne-Valensole basin suggest the region was affected by the Middle Miocene climate dynamics similarly as the global oceans, in addition to significant and rapid changes in temperature and seasonality as a result of a reorganization in atmospheric circulation.

## 6. Conclusions

Our Early to Middle Miocene terrestrial T($\Delta_{47}$), $\delta^{13}C_{carb}$, $\delta^{18}O_{carb}$, and $\delta^{18}O_w$ record for the Northern Mediterranean Digne-Valensole basin (SE France) shows that:

1) An abrupt decrease in $\Delta_{47}$ temperature (17.5 ºC within 600 ka) occurs at 13.8 Ma, which we attribute to post-MMCT cooling whose timing is in agreement with global marine (Holbourn et al., 2022; Westerhold et al., 2020) and Mediterranean (Abels et al., 2005) records. The decrease in $\Delta_{47}$ temperatures associated with a major decrease in $\delta^{13}C_{carb}$ (-4.0 %), $\delta^{18}O_{carb}$ (-2.5 %), and $\delta^{18}O_w$ (-5.0 %) values within less than 600 ka may be best explained by a large seasonality component and overall cooling on the order of 3-4 °C.

2) During the MCO, DVB $\Delta_{47}$-temperatures are higher than other reconstructed European mid-latitudinal average warm month temperatures (Ivanov and Böhme, 2011; Mosbrugger et al., 2005).

3) Similar to the observations from the Swiss NAFB record (Methner et al., 2020), we suggest that variations in $\Delta_{47}$-temperatures record the combined effect of global warming and cooling and changes in the annual distribution of rainfall, which affects the timing of pedogenic carbonate formation. The latter would result from the reorganization of atmospheric circulation in the circum-Alpine region.

4) We suggest that a proto-Mediterranean climate characterized by hot and dry summers already existed during certain time intervals of the MCO.

5) We identify an increase in soil $\Delta_{47}$-temperatures at 15 Ma, which coincides with warming trends observed in global marine data (Cramer et al., 2011; Westerhold et al., 2020).





6) We observe a cooling phase in our $\Delta_{47}$-temperatures between circa 16.7 and 15.6 Ma which coincides with cooling in global sea surface and bottom water temperatures (Cramer et al., 2011; Lear et al., 2015; Super et al., 2018).

7) The coolest recorded temperature during the MCO in our record coincides with the $\delta^{18}O_{benthic}$ excursion Mi3a (14.8 Ma) and associated transient glacial expansion (Miller et al., 2020).

**Data availability.** All supporting datasets are available as Supplementary information files that will be freely accessible upon publication.

**Authors contribution.** AB: Writing – original draft. AB, MJMM, AM, IC, MB, DH, KM and MT: Writing – review & editing. AB, MJMM, KM and AM: Conceptualization. AB and IC: Investigation. AB: Visualization. MJMM and AM: Funding acquisition. MJMM, AM and IC: Supervision. MB: Software. MB and MT: Formal analysis. JF: Methodology, Resources.

**Declaration of competing interest.** The authors declare that they have no known competing financial interests
or personal relationships that could have appeared to influence the work reported in this paper.

**Acknowledgements.** AM and MJM acknowledge funding through grants MU2845/7-1 and ME5579/1-1 of the German Science Foundation. (DFG) in the form of Special Priority Program 2020 'Mountain-Building in Four-Dimensions' (DFG-SPP 4D MB). We thank Sven Hofmann for assistance with clumped isotope analyses and Alexis Licht for field assistance.

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
