# Peer review of "Middle Miocene climate evolution in the Northern Mediterranean region (Digne-Valensole basin, SE France)"

_EGUsphere, 2024_

## Author Response (AR1)

Referee #1

The authors investigate the Middle Miocene Climatic Transition in a Mediterranean Alpine foreland basin. This paleoclimate record provides a land-based perspective of climate change during the transition from globally warm to globally cool conditions. The authors use clumped isotope temperatures from pedogenic carbonate to infer patterns in secular temperature change and as a record of hydrologic change. They interpret a changing seasonal bias in their pedogenic carbonates and suggest it relates to the onset of Mediterranean climates.

The authors present a creative yet measured interpretation of their data. I recommend minor revisions according to the suggestions below.

We thank the reviewer for their constructive comments and suggestions. Our replies can be found below (in black).

Reviewer #2 caught a mistake in the calcite-water oxygen isotope fractionation-temperature relationship that we used. We have corrected this in tables and figures (see below) and we would like to point out that the changes are minor and have not affected our interpretation.

During the revision process we updated 3 citations and phrasing of the clumped isotope thermometry methodology in the main text of the manuscript and the supplementary material.

Line numbers in our replies refer to the line numbers in the revised manuscript.

Line 180 - is this sampling depth below the paleo ground surface? How did you determine the top of the paleosol? Or do you mean you excavated below 50 cm from the modern surface to avoid contemporary overprinting?

Nodules were sampled by Bialkowski (2002) at depths between 30 cm and 1 m below the preserved soil top, which corresponds to a truncation surface, ensuring that the original depth of the nodules was at least 50 cm. They also excavated the sampling locations to avoid contemporary overprinting.

The identification of the top of the paleosol was rendered possible due to distinct colouring and/or lithostratigraphic characteristics of the horizons (see Cojan and Gillot, 2022).

We changed the text accordingly, line 181: "Nodules were sampled at depths between 50 cm and 100 cm below the preserved soil top, which at times corresponds to a truncation surface. Therefore the formation depth of the carbonate nodules was at least 50 cm (Bialkowski, 2002)."

How much time do you think each pedogenic nodule represents? Do you think it is smaller than the error from your age model?

The timing of formation of pedogenic carbonate nodules is debated. While Zamanian et al. (2016) propose that nodules form within decades, other studies suggest a development over hundreds to thousands of years (Gile et al., 1966; Kelson et al., 2020). Therefore, the uncertainty in our age model (ranging from 100 ka to 1 Ma) is larger than the time interval represented by the formation individual carbonate nodules in the paleosols.

We added a sentence to the manuscript that clearly states that the uncertainty in our age model is (much) larger than the time interval represented by the formation of the individual carbonate nodules in the paleosols, line 207: "Age uncertainties in the DVB range from 100 ka to 1 Ma (Fig. 3; Bialkowski, 2002). The age uncertainties in the DVB are constrained by the correlation of the DVB biostratigraphy with the biostratigraphy of the well-correlated magnetostratigraphic NAFB and the well-described biostratigraphy (MN zones) of the Aquitaine basin (Bialkowski, 2002). Given that individual carbonate nodules in the paleosols may form over hundreds to thousands of years (Gile et al., 1966; Kelson et al., 2020), the uncertainty in the age model is larger than the time integrated by the individual carbonate nodules."

Line 280 - It would be nice to see isotopic measurements of the sparry veins to reassure the reader that the rest of the data that is 'pristine' is pristine.

We observe no significant differences in the stable isotopic data between the bulk nodule (98GR22) and major sparry veins (98GR22v, see Table S1). The fact that the results of the nodule with calcite veins fall in the range of $\delta^{18}O$ and $\delta^{13}C$ values observed in micrites is not uncommon (Garzione et al., 2004). In the methodology of nodules selection, Bialkowski (2002) described that after thin section analyses, they observed only rare cases of recrystallization in the micritic carbonate nodules. Additional work (e.g. petrographic, LA-ICPMS, SEM) on this nodule would go beyond the scope of our paper.

We added the isotopic data of the sparry veins to the Table S1.

Additionally, we noticed a mistake in the mineralogical description of the nodule with sparry vein in Figure S2d and corrected it.

Can you offer some kind of contextual information on why there would be local fluid flow affecting just this one sample? Was this collected in a disparate location?

This is a very valid question. The nodule was not collected in a disparate location. We certainly did not want to suggest that this is the only sample affected by local fluid flow. There may very well be other pedogenic carbonates with sparry calcite veins, but we unfortunately selected this one for clumped isotope analysis.

We would further like to point out that we removed the sample based on its clumped isotope temperature alone, which is slightly higher than plausible on the Earth's surface. However, its $\delta^{18}O$ and $\delta^{13}C$ values fall within the range of $\delta^{18}O$ and $\delta^{13}C$ values observed in the other micritic nodules of this section. As a note of caution, we rejected this sample from further interpretation. However, the inclusion of the sample in our analysis would not change our interpretation and conclusions.

We incorporated the information above in the manuscript, line 248: "This sample yields a clumped isotope temperature (T($\Delta_{47}$) = 39.2 °C), which is slightly higher than plausible on the Earth's surface (see Li et al., 2024). Its $\delta^{18}O_{carb}$ and $\delta^{13}C_{carb}$ values fall within the range of $\delta^{18}O_{carb}$ and $\delta^{13}C_{carb}$ values observed in the other micritic nodules of this section. However, as a note of caution, we rejected this sample from further interpretation, even though the inclusion of the sample in our analysis would not change our interpretation and conclusions."

Consider citing Li et al. GCA (2024) (https://doi.org/10.1016/j.gca.2024.04.009) in this section.

We cited Li et al. (2024) line 249.

Line 302 - at what depth?

Soil temperatures are typically higher than air temperatures due to solar radiation heating. Soil temperature varies over time and decreases exponentially with increasing soil depth (Quade et al., 2013). At shallow depths (< 30 cm), soil temperature exhibits significant daily fluctuations, but these diurnal variations diminish with depth. Beyond 300 cm, soil temperature becomes nearly constant throughout the year, converging toward the mean annual temperature (Quade et al., 2013).

We included this information in the discussion, line 314: "Soil temperatures are typically higher than air temperatures due to solar radiation heating (Molnar, 2022; Quade et al., 2013). At shallow depths (< 30 cm), soil temperature exhibits significant daily fluctuations, but these diurnal variations diminish with depth. Beyond 300 cm, soil temperature becomes nearly constant throughout the year, converging toward the MAT (Molnar, 2022; Quade et al., 2013; Sheldon and Tabor, 2009)."

Line 304 - does the shift in seasonality have to be instantaneous? Could it be gradual based on the uncertainties in your age model and in your sampling resolution?

The shift in seasonality does not have to be instantaneous, but we try to give the reader a feel for the extremes that would be possible by a change in rainfall seasonality under modern climate. Hence the word 'hypothetical' in the sentence.

**Figure 4C: from where did you estimate the monthly threshold for precipitation amount preventing pedogenic carbonate?**

The parameters in Figs. 4A to 4C are based on real locations in France and Hungary. As such, it is known that no pedogenic carbonates are formed in Brest (France) under modern climatic conditions and the monthly threshold is irrelevant

We added information in the caption of Figure 4: "Present-day climate conditions for three location in Europe: (A.B) Aix en Provence and Brest (France) and (C) Debrecen (Hungary). Variations in seasonality of precipitation may drive changes in pedogenic carbonate formation temperatures. Depending on the seasonality of pedogenic carbonate formation, soil temperatures reconstructed from pedogenic carbonate should reflect (A) warm (> MAT) or (B) cool (≤ MAT) season temperatures. In (C), conditions for the formation of pedogenic carbonate are not met because of the lack of contrasting rainfall seasonality and temperature throughout the year, hence no pedogenic carbonate is formed at this location. The $\delta^{18}O_w$ values are retrieved from the GNIP stations near the locations. (D) Influence (cross: large, circle: moderate) of the interplay between soil temperature and soil water deficit on the formation of pedogenic carbonate. Temperature dominated (A), water deficit dominated (B), and no pedogenic carbonate formation (C) scenarios. "

**Line 350: What are the d18O values of these moisture sources today? You could consider adding the seasonal d18O signatures to Figure 4.**

We thank the reviewer for this suggestion. The modern long-term $\delta^{18}O$ values of precipitation in the Digne-Valensole region (GNIP station located in Draix) are much lower, averaging -8.35 ‰, than our reconstructed Miocene $\delta^{18}O$ data. The modern precipitation $\delta^{18}O$ values range from -11.27 ‰ (winter) to -4.77 ‰ (summer).

We added the seasonal $\delta^{18}O$ signatures to each scenario in Figure 4.

**How would you expect the d13C values to vary with the changing seasonal bias in pedogenic carbonate formation?**

We thank the reviewer for this interesting comment. The primary factors influencing $\delta^{13}C$ values of pedogenic carbonates are the carbon isotopic composition of soil water $CO_2$, which is largely controlled by vegetation type (C3 vs. C4 biomes), biological activity, and environmental factors (e.g., temperature, moisture, vegetation), as well as atmospheric $CO_2$ to some extent (Cerling, 1984). We rule out a possible early shift in vegetation from C3- to C4-dominated biomes as (1) this would occur couple of million years earlier than mostly observed in the region (Strömberg, 2011) and (2) the observed $\delta^{13}C$ values of -6 to -9 ‰ fall well into the range of C3-dominated biomes. We suggest that lower amount of rainfall would reduce the depth of carbonate precipitation, as well as soil productivity and

respiration, which results in more enriched $\delta^{13}C$ values (Cerling 1984; Stevenson et al., 2004). Regarding the temperature factor, we suggest $\delta^{13}C$ values of pedogenic carbonates would decrease with rising temperatures, primarily due to increased biological activity and $CO_2$ production in warmer conditions. However, isotopic fractionation during carbonate precipitation moderates this relationship, making it a complex interaction between biological and geochemical processes. We would therefore expect lower $\delta^{13}C$ values during time periods associated with cooler temperatures and higher precipitation amount. This agrees with the significant decrease in $\delta^{13}C$ values during the transition associated with the MMCT time interval in our record (Fig. 3).

We incorporated a short paragraph in the manuscript that details the drivers of variations in $\delta^{13}C$ values, line 338: "Shifts in the seasonality of pedogenic carbonate formation could potentially be associated with variations in $\delta^{13}C_{carb}$ values. We suggest that $\delta^{13}C$ of pedogenic carbonates would decrease with rising temperatures, primarily due to increased biological activity and $CO_2$ production. However, isotopic fractionation during carbonate precipitation moderates this relationship, making it a complex interaction between biological and geochemical processes. However, we would expect lower $\delta^{13}C_{carb}$ values during time periods associated with cooler temperatures and high precipitation amount. This agrees with the significantly low $\delta^{13}C_{carb}$ values during the MMCT in our record (Fig. 3)."

Line 403: this is a nice, clear statement that summarizes your interpretation.

We thank the reviewer for this acknowledgment.

Consider combining Figure 5 and 6.

We thank the reviewer for this suggestion. We prefer not to combine Figures 5 and 6, as one focuses on regional paleoclimate settings from terrestrial proxies while the other addresses a broader global context.

Line 426: This discussion and your bulleted conclusions are difficult to match with Figure 6 because you do not have 0.1 Ma's marked on that figure. Revise the figure axes or clarify the samples you are discussing in an alternate way.

We adapted Figure 6 accordingly and added indication for 0.1 Ma's.

Line 410 and elsewhere in this section: Based on the prior section, I thought that the temperature variability in the MCO that you observed on land is primarily related to changing seasonality of pedogenic carbonate formation rather than a secular change in temperatures. I suspect that variability in temperatures in the marine realm is caused by a different mechanism. Perhaps you could discuss why you might expect

to see variability in measured temperatures across these two regions even though the mechanisms for variability differ.

Global climate change will not only lead to an increase or decrease of marine and continental temperatures, but may also significantly impact atmospheric circulation, which leads to altered weather patterns, ocean-atmosphere interactions (Holbourn et al., 2005; 2007; Methner et al., 2020) similar to changes observed during anthropogenic warming. Changes in temperature gradients can weaken or shift trade winds, disrupting weather systems and ocean currents (Comas-Bru et al., 2016). Because the rapid and large changes in measured soil carbonate T($\Delta_{47}$) during the Middle Miocene are too large to be explained by regional/global cooling or warming, we suggest that the changes in T($\Delta_{47}$) are resulting from the combined effect of regional/global cooling or warming and changes in seasonality of rainfall. However, it is challenging to decompose the temperature shift into a seasonal component and a global climate component. However, both must be at work simultaneously.

We pointed this out more clearly in the manuscript line 430: "Global climate change would not only lead to a decrease (or increase) of marine and continental temperatures, but may also significantly impact atmospheric circulation resulting in altered weather patterns and ocean-atmosphere interactions (Holbourn et al., 2005; 2007; Methner et al., 2020) similar to changes observed during anthropogenic warming. Changes in temperature gradients can weaken or shift trade winds, disrupting weather systems and ocean currents (Comas-Bru et al., 2016). Throughout the MCO and MMCT, our northern Mediterranean terrestrial soil $\Delta_{47}$-temperatures show significant similarities with marine temperature records (Fig. 6). Because the rapid and large changes in measured soil carbonate T($\Delta_{47}$) during the Middle Miocene are too large to be explained by regional/global cooling (or warming) alone, we suggest that the changes in soil $\Delta_{47}$-temperatures are resulting from the combined effect of changes in seasonality of rainfall and regional/global cooling (or warming)."

Line 445: Where does the estimate of cooling come from? I don't think you explained which temperature values you are comparing to arrive at the estimated 3-4 °C.

Under modern temperature seasonality, soil carbonate formation temperatures can change up to 17 ºC depending on the carbonate precipitation season. Accordingly, we assume that the temperature change of ca. 14 ºC observed between 14.9 Ma and 14.2 Ma reflects only a shift in seasonality without any major regional temperature change. We imply therefore that any fluctuations in carbonate formation temperatures larger than 14 ºC may result of additional regional or global warming or cooling.

We appreciate the comment of the reviewer and we clarified this point in the text, line 417: "Assuming that the temperature change of ca. 14 ºC (20.6 ºC to 34.7 ºC) that is observed between 14.9 Ma and 14.2 Ma reflects only the change in carbonate

formation season without any major regional temperature change, we posit that any fluctuations in carbonate formation temperatures larger than 14 ± 2 ºC must be the result of additional global warming or cooling. We therefore propose that the remaining 3-4 °C temperature decrease after 13.8 Ma reflects the effect of regional post-MMCT cooling."

We suggest in line 453 that a climate similar to the present-day Mediterranean-type climate (characterized by hot and dry summers) occurred during certain time intervals of the MCO, given that high carbonate formation temperatures must reflect carbonate formation during the summer. More specifically, we refer to the relatively high T($\Delta_{47}$) (> 30ºC) and high $\delta^{18}O_w$ values (> -2 ‰) at 16.5 Ma and at 14.2–13.8 Ma.

We clarified this in the text, line 365: "We hypothesize that a transition to a climate similar to the present-day Mediterranean-type climate ('proto-Mediterranean climate'), characterized by hot and dry summers (Fig. 4A), occurred during the Middle Miocene (at ca. 16.5 Ma and 14.2–13.8 Ma; Fig. 5)."

And line 373: "We therefore propose that high T($\Delta_{47}$) (> 30 ºC) and high $\delta^{18}O_w$ values (> -2 ‰), during the MCO and MMCT resulted from a Mediterranean moisture source ('proto-Mediterranean climate'), whereas lower $\Delta_{47}$-temperatures and lower $\delta^{18}O_w$ values (< -2 ‰) reflect predominance of a North Atlantic moisture source."

We also incorporated indications in Figure 5.

Line 457: Based on Fig 6, it looks like there is a slight peak in BWT in the interval that you are describing. I'm not sure which wiggle you are matching to - again, adding 0.1 Ma to FIg 6 would help, and/or label these tie points on the graph.

We appreciate the comment of the reviewer and modified the figure accordingly.

Referee #2

The authors presented the stable and clumped isotope results of soil carbonates formed during the middle Miocene (23–13 Ma) from the northern Mediterranean Alpine foreland basin in France. And they proposed that the clumped isotope derived temperature reflected the combined effects of long-term regional temperature and carbonate formation seasonality changes. Except for the comments from referee 1, I have following concerns with the current version.

1. According to the visual comparison with Figure 4 in Bialkowski et al. (2006), it seems there are large changes in the age model of this work. For example, the 98GR24 sample may correspond to 16-17 Ma in Figure 4 of Bialkowski et al. (2006). If so, the authors should give more details and the reasons for such

changes. Furthermore, what is the underlying physical basis for correlating the carbon isotope of soil carbonates with the marine data in Figure S4?

We thank the reviewer for this comment. We updated the Supplementary Material and included a comparison of the age models in Figure S4.

First, we would like to point out that there are significant changes between the age models because of the use of different time scales. As explained in S3.2, we follow the same protocol as Bialkowski et al. (2006) (who used the time scale of Berggren et al., 1995) but we adopt and update the age model to the benthic curve of Westerhold et al. (2020) and accordingly to the more recent astronomically tuned time scale of Hilgen et al. (2012).

Second, there are only minor changes between the age models in the interval that include sample 98GR24. The most significant differences in the two age models occur in the interval from 19 to 22 Ma, where tie points are scarce.

Thirs, the age models pf Bialkowski et al. (2006) and this study are – at first order – based on the correlation of micromammal assemblages, pollen, and dynocysts to the time scale of Hilgen et al. (2012; see Fig. S4 for correlations). The $\delta^{13}$C values of the pedogenic carbonates were used as a second order correlation.

A key assumption in using pedogenic carbonate nodules for carbon isotope chemostratigrahpy is that their $\delta^{13}$C variation is mainly governed by changes in global atmospheric and oceanic $\delta^{13}$C values, allowing them to be pattern-matched with $\delta^{13}$C values from marine carbonates (Cerling, 1984; Bataille et al., 2016). In their study, Bialkowski et al. (2006) describe how they employ carbon isotope chemostratigraphy in combination with sedimentologic and biostratigraphic data to establish an age-model for the Digne-Valensole section. It is beyond the scope of this paper to discuss the quality and limitations of the age model developed by their work using pedogenic carbonate nodules as a substrate for carbon isotope chemostratigraphy.

1. Studies have suggested that there were large changes in the illuvial depth of CaCO3 during the Quaternary (Zhao, 2004; Meng et al. 2015). From Line 130-135, the studied region may experience large changes in precipitation during the MCO, and this may induce large changes in soil carbonate formation depth. Figure 2B also show several meters of carbonate nodules, implying large formation depth. According to the study of Quade et al. (2013), this could also reasonably explain the large changes in the clumped isotope temperature of soil carbonates, which is ignored in this work.

Zhao, J. (2004), The new basic theory on Quaternary environmental research, J. Geogr. Sci., 14(2), 242–250, doi:10.1007/BF02837540.

Meng, X., L. Liu, W. Balsam, S. Li, T. He, J. Chen, and J. Ji (2015), Dolomite abundance in Chinese loess deposits: A new proxy of monsoon precipitation intensity, Geophys. Res. Lett., 42, 10,391–10,398, doi:10.1002/2015GL066681.

We thank the reviewer for this comment.

In most of the studied paleosols, the accumulation horizon of the carbonates was preserved which allows the identification of the paleosurface of the soil. The carbonate nodules were sampled at 50-100 cm below the preserved soil top (Cojan et al., 2013). The large variations in $T(\Delta_{47})$ (≥ 10 ºC) we attribute to changes in seasonality (coupled with global warming/cooling) are too substantial to be explained by changes in depth, as suggested by the results of Kelson et al. (2020) and Molnar (2022).

While this mechanism is not explicitly discussed in the current version of the manuscript, it is an important factor and we will therefore modify the text accordingly.

We added the following sentence line 293: "Owing that the sampling depth of carbonate nodules does not exceed 1 m below the preserved surface of the paleosol, we rule out the hypothesis that changes in depth associated with fluctuations in precipitation amount significantly affect the measured $T(\Delta_{47})$ (Kelson et al., 2020; Molnar, 2022)."

1. Line 445, the analytical error (1SE) for the clumped isotope is on the order of 3-4 ℃. So, it may be difficult to differentiate the overall cooling trend from large seasonality component.

We acknowledge that the analytical error (1SE) for the clumped isotope temperatures is on the order of 3–4 ºC, which indeed makes it challenging to disentangle an overall cooling trend from a large seasonality component. However, the concurrent major decreases in $\delta^{13}C_c$ (-4.0 ‰), $\delta^{18}O_c$ (-2.5 ‰) – which are independent to clumped isotope thermometry – in addition to $\delta^{18}O_w$ (-5.0 ‰) within a relatively short timescale (< 600 ka) suggest significant climatic changes occurred. While part of the $\Delta_{47}$ signal may reflect a rainfall seasonality effect, the magnitude and direction of the isotopic changes (e.g., difference of 5 ‰) collectively support the likelihood of ambient regional warming and cooling, even if the precise extent remains with the range of uncertainty.

As suggested by referee #2, we clarified this point in the text to better reflect the limitations and the interpretation, line 423: "While the analytical error (1SE) for the clumped isotope temperatures is on the order of a few degrees, the concurrent major decreases in $\delta^{13}C_{carb}$, $\delta^{18}O_{carb}$, and $\delta^{18}O_w$ values within a relatively short timescale (< 600 ka) suggest considerable climatic changes occurred."

1. Line 65, Kim & O'Neil 1997 is more appropriate here and other related parts in the paper.

We thank the reviewer for catching this and we regret that this has happened. We have replaced Kim et al. (2007) by Kim and O'Neil (1997) throughout the manuscript. Accordingly, we have recalculated all $\delta^{18}O_w$ values in Table 1, Table S1, as well as implemented the adjusted values in Figure 3 and Figure S3. The adjusted values are, on average, 0.2 ‰ higher, which is smaller than the symbol size in Figure 3

(right panel). Given the marginal change in $\delta^{18}O_w$ values, our interpretation of the data is not affected.